# sRAGE as a Prognostic Biomarker in ARDS: Insights from a Clinical Cohort Study

**DOI:** 10.3390/medicina61020229

**Published:** 2025-01-27

**Authors:** Ana Andrijevic, Uros Batranovic, Djordje Nedeljkov, Srdjan Gavrilovic, Vladimir Carapic, Svetislava Milic, Jovan Matijasevic, Ilija Andrijevic

**Affiliations:** 1Medical Faculty, University of Novi Sad, 21000 Novi Sad, Serbia; srdjan.gavrilovic@mf.uns.ac.rs (S.G.); vladimir.carapic@mf.uns.ac.rs (V.C.); jovan.matijasevic@mf.uns.ac.rs (J.M.); ilija.andrijevic@mf.uns.ac.rs (I.A.); 2Intensive Care Unit, Institute for Pulmonary Diseases of Vojvodina, 21204 Sremska Kamenica, Serbia; batranovic@gmail.com (U.B.); nedeljkovdjordje@gmail.com (D.N.); svetislava.milic@yahoo.com (S.M.)

**Keywords:** ARDS, sRAGE, biomarkers, ICU, mortality

## Abstract

*Background and Objectives*: Acute respiratory distress syndrome (ARDS) is a severe form of acute lung injury with high mortality, characterized by hypoxemic respiratory failure and diffuse lung damage. Despite advancements in care, no definitive biomarkers have been established for ARDS diagnosis and prognostic stratification. Soluble receptor for advanced glycation end-products (sRAGE), a marker of alveolar epithelial injury, has shown promise as a prognostic indicator in ARDS. This study evaluates sRAGE’s utility in predicting 28-day mortality. *Materials and Methods*: A retrospective cohort study was conducted at a tertiary care ICU in Serbia from January 2021 to June 2023. Adult patients meeting the Berlin definition of ARDS were included. Exclusion criteria included pre-existing chronic respiratory diseases and prolonged mechanical ventilation before diagnosis. Serum sRAGE levels were measured within 48 h of ARDS diagnosis using enzyme-linked immunosorbent assay (ELISA). Clinical severity scores, laboratory markers, and ventilatory parameters were recorded. Logistic regression and survival analyses were used to assess the prognostic value of sRAGE for 28-day mortality. *Results*: A cohort of 121 patients (mean age 55.5 years; 63.6% male) was analyzed. Non-survivors exhibited higher median sRAGE levels than survivors (5852 vs. 4479 pg/mL, *p* = 0.084). The optimal sRAGE cut-off for predicting mortality was >16,500 pg/mL (sensitivity 30.4%, specificity 86.9%). Elevated sRAGE levels were associated with greater disease severity and an increased risk of 28-day mortality in ARDS patients, highlighting its potential as a prognostic biomarker. The main findings, while indicative of a trend toward higher sRAGE levels in non-survivors, did not reach statistical significance. *Conclusions*: The main findings, while indicative of a trend toward higher sRAGE levels in non-survivors, did not reach statistical significance (*p* = 0.084). sRAGE demonstrates potential as a prognostic biomarker in ARDS and has moderate correlation with 28-day mortality. Integrating sRAGE with other biomarkers could enhance risk stratification and guide therapeutic decisions. The retrospective design limits the ability to establish causation, underscoring the need for multicenter prospective studies.

## 1. Introduction

Acute lung injury manifests in over 10% of patients admitted to intensive care units (ICUs), with acute respiratory distress syndrome (ARDS) being its most severe form, associated with a mortality rate of approximately 40% [1]. According to the Berlin Definition of 2012, ARDS is defined by specific diagnostic criteria, including acute onset or worsening of hypoxemic respiratory failure within 7 days of the estimated predisposing factor, radiographic evidence of bilateral opacities within the lung parenchyma unexplained by pleural effusion, lobar/lung collapse, nodules, or other causes of respiratory insufficiency that are not attributable to cardiac failure or fluid overload [2]. Stratification of ARDS severity is based on the ratio of arterial oxygen partial pressure to the fraction of inspired oxygen (PaO_2_/FiO_2_), and ARDS is classified into three groups: mild with a PaO_2_/FiO_2_ ≤ 300, moderate with PaO_2_/FiO_2_ ≤ 200, and severe with PaO_2_/FiO_2_ ≤ 100 [3].

ARDS can be triggered by numerous causes, with effective management of the underlying cause being critical for improving patient outcomes [4]. The syndrome is heterogeneous, reflecting an acute inflammatory response within the lung parenchyma, characterized by a pathognomonic triad: increased pulmonary vascular permeability, increased lung weight, and reduction in aerated lung tissue [2]. Pneumonia is the predominant cause of ARDS, underscoring the importance of systematic microbiological evaluation for potential pathogens in the diagnostic workflow. Identifying the pathogenic mechanism underlying lung injury is crucial for optimizing both etiologic and supportive therapy [5]. Besides pneumonia, ARDS may arise from extrapulmonary sepsis, aspiration of gastric contents, pulmonary contusion, major trauma, burns, pancreatitis, inhalational injuries, non-cardiogenic shock, multiple transfusions or transfusion-related lung injury, pulmonary vasculitis, and drowning [2].

As a significant global health challenge, ARDS exhibits variable incidence across regions, with rates in Europe reported to be ten times lower than in the United States. The studies have shown an ICU mortality rate as high as 35.3% and an overall hospital mortality rate of approximately 40% [5]. Also, the criticality of timely ARDS diagnosis is of great importance as the condition remains unrecognized in 40% of cases, with recognition rates of 51% in moderate and 79% in severe cases [6].

From both a pathophysiological and clinical perspective, ARDS comprises two overlapping stages: the initial exudative and the subsequent fibroproliferative phase [7]. The exudative phase, occurring within 48–72 h of injury onset, is marked by endothelial and epithelial cell damage, an inflammatory cascade, and increased vascular permeability [1].

Current diagnostic and stratification approaches for ARDS are based on a combination of clinical and pathophysiological assessments, as no definitive biomarker has been identified yet to facilitate ARDS diagnosis and treatment [8]. Among the numerous biomarkers studied in the pathophysiology of ARDS, soluble receptor for advanced glycation end-products (sRAGE) has emerged as a potential key mediator, bridging the inflammatory and alveolar epithelial injury pathways characteristic of the syndrome. The receptor for advanced glycation end-products (RAGE) was initially identified in respiratory samples in 1992 [9]. RAGE is a transmembrane protein belonging to the immunoglobulin superfamily and is a key component of cell surface molecules. As a glycosylated protein with a molecular mass of 50–55 kDa, RAGE comprises an extracellular domain, a hydrophobic transmembrane region, and a cytoplasmic tail [10]. RAGE is expressed in multiple isoforms, facilitating its binding to a wide array of endogenous extracellular ligands and intracellular effectors. Ligand binding to the extracellular domain of RAGE activates a complex intracellular signaling cascade, resulting in reactive oxygen species production, immunoinflammatory responses, cellular proliferation, or apoptosis [11]. Cleavage of the extracellular domain produces a soluble form of RAGE, termed sRAGE, with a molecular mass of 35 kDa, detectable in plasma or serum [12]. In the context of ARDS, sRAGE is recognized as a marker of type I pneumocyte epithelial injury [13].

Jabaudon et al. [14] conducted a comprehensive analysis demonstrating that elevated baseline plasma sRAGE levels are independently associated with increased 90-day mortality in ARDS patients. Their findings highlighted the utility of sRAGE as an indicator of disease severity and a predictor of clinical outcomes, reinforcing its potential as a key component in risk stratification and management protocols for ARDS. Notably, their work emphasized the pathophysiological link between alveolar epithelial damage and the systemic inflammatory response, which contributes to the progression of ARDS [14]. Complementing this, Blondonnet et al. explored the pathophysiological significance of sRAGE and its related ligands in ARDS, identifying it as a valuable marker for assessing lung injury severity. Their study underscored the biomarker’s dual role in diagnosing the extent of epithelial damage and in guiding therapeutic interventions, particularly in heterogenous clinical settings. They also compared sRAGE with other biomarkers, such as surfactant protein-D (SP-D), another marker of epithelial injury, noting that while SP-D is indicative of type II alveolar cell damage, sRAGE uniquely reflects type I alveolar epithelial injury, offering complementary insights into the mechanisms underlying ARDS. The study proposed that sRAGE could serve not only as a prognostic tool but also to tailor individualized treatment strategies, aligning with the broader goal of personalized medicine in critical care. These findings collectively support the hypothesis that sRAGE may provide critical insights into ARDS pathophysiology and prognosis [7].

The prognostic value of serum biomarkers in ARDS offers clinical potential, as these biomarkers may provide insights into disease severity, progression, and treatment response. Biomarkers reflecting epithelial and endothelial damage might be correlated with worse outcomes in ARDS patients, highlighting their potential for prognostic stratification [1].

Identifying a serum biomarker that correlates with clinical presentation and laboratory findings could significantly aid in disease diagnosis and enable clinicians to stratify ARDS severity more accurately, as well as assisting in sequencing interventions. Incorporating the timing and sequencing of interventions into ARDS management protocols is crucial to standardizing care, improving outcome consistency, and effectively stratifying disease severity [15].

## 2. Materials and Methods

### 2.1. Study Design

This research is a retrospective, observational cohort study design conducted in the Intensive Care Unit (ICU) of the Institute for Pulmonary Diseases of Vojvodina, Serbia, a university hospital center specializing in critical care. The research period spanned from 1 January 2021 to 30 June 2023. Due to the retrospective nature of the study, data were extracted from pre-existing medical records and laboratory databases within the specified timeframe.

This study aims to evaluate the prognostic value of sRAGE, a novel biomarker reflecting type I alveolar epithelial injury, in predicting 28-day mortality in ARDS patients. This work proposes a specific cut-off for sRAGE levels and examines its correlation with clinical severity scores and outcomes, providing new insights into its potential clinical application.

### 2.2. Study Population

Inclusion Criteria: The study included adult patients (≥18 years) diagnosed with ARDS according to the Berlin definition.

Exclusion Criteria: Patients were excluded if they met any of the following criteria:-Pre-existing chronic respiratory diseases that could alter sRAGE levels, such as pulmonary fibrosis, which could confound biomarker interpretation.-Receiving mechanical ventilation for more than 48 h prior to blood sampling. This exclusion ensures the values represent the proliferative phase of ARDS and reduces the potential variability in sRAGE levels that prolonged ventilation might introduce.

### 2.3. Data Collection

Data were extracted from patient records and laboratory databases from the hospital’s electronic charts and organized into the following categories:
Baseline Data
-Demographics: age and gender;-Comorbidities: each individually recorded and using the Charlson comorbidity index;-Clinical severity: quantified using APACHE II, SAPS2, and SOFA scores calculated upon ICU admission;-Laboratory tests: baseline inflammatory markers (C reactive protein, procalcitonin), lactate;-Mechanical ventilation parameters: Tidal volume (mL), PEEP (cmH_2_O), FiO_2_, plateau pressure (cmH_2_O), driving pressure (cmH_2_O), static compliance (mL/cmH_2_O);-Arterial blood gas parameters: PaO_2_/FiO_2_, pCO_2_ (kPa), pH.
sRAGE Measurements
-Sample Collection: Serum samples for sRAGE measurement were obtained within the first 48 h of ARDS diagnosis to capture an accurate baseline biomarker level;-Biomarker Assay: sRAGE levels were measured using a standardized enzyme-linked immunosorbent assay (ELISA) kit, following protocols specified by the manufacturer to ensure reproducibility and accuracy;-Processing and Storage: All serum samples were stored at −80 °C until batch processing, minimizing inter-assay variability and maintaining sample integrity for accurate biomarker analysis.
Outcome Measures
-Primary Outcome: 28-day mortality;-Secondary Outcomes: length of ICU stay and ventilator-free days within the first 28 days.


### 2.4. Statistical Analysis

Summaries of patient characteristics, sRAGE levels, and clinical outcomes were reported as means with standard deviations or medians with interquartile ranges, depending on the data distribution, for continuous variables, and as frequencies and percentages for categorical variables. Continuous variables were also presented using density plot charts. Patients were stratified by sRAGE quartiles or other relevant clinical categories. Mortality and other categorical outcomes were analyzed using chi-square or Fisher’s exact tests, depending on sample size. Continuous outcomes were evaluated using *t*-tests or Mann–Whitney U tests, depending on the distribution of the data. Logistic regression models were used to assess the predictive value of sRAGE for mortality, adjusting for potential confounders such as age, gender, and comorbidities. Kaplan–Meier survival curves were constructed, and log-rank tests were applied to compare survival times across sRAGE quartiles. Cox proportional hazards models might further evaluate the association between sRAGE levels and time-to-event outcomes, adjusting for baseline risk factors such as age, comorbidities, and gender. A *p*-value of 0.05 or lower was considered statistically significant.

The sample size was based on an assumed in-hospital rate of death of 50% and expected sRAGE standard deviation of 4000. Thus, an enrollment of 126 would have a power of 80% (at a two-sided alpha level of 0.05) to detect difference in sRAGE means of 2000 between groups. While limited, the sample size reflects the constraints of a single-center retrospective study. We acknowledge that incorporating dynamic biomarker data could provide deeper insights into the prognostic role of sRAGE. Additionally, the lack of investigation into sRAGE’s performance in predicting disease progression represents a limitation of our study.

### 2.5. Ethical Considerations

This study was reviewed and approved by the Institutional Review Board of the Institute for Pulmonary Diseases of Vojvodina, ensuring adherence to ethical standards for retrospective data collection. Given the retrospective nature of the study, a waiver of informed consent was granted due to anonymized data collection. All patient data were de-identified to maintain confidentiality in accordance with relevant data protection laws and institutional policies.

## 3. Results

A total of 121 patients with ARDS were included in the study. The mean age of the patient cohort was 55.5 ± 15.0 years, with ages ranging from 20 to 83 years. A total of 77 patients were male (63.6%). The majority presented with severe ARDS (47.9%), followed by moderate ARDS (42.1%) and mild ARDS (9.9%). Demographic data and baseline data are presented in Table 1.

Within the initial 24 h following ICU admission, 28% of patients required vasopressor support. During this same period, acute renal failure requiring renal replacement therapy developed in 22 patients (18.2%). Venovenous extracorporeal membrane oxygenation (VV ECMO) was conducted in 10 patients (8.3%).

The predominant etiological factor in cases of infectious ARDS was viral, accounting for 53.7% of patients, followed by bacterial infections, identified in 17.4% of cases. One patient had a co-infection involving both viral and bacterial pathogens, while no specific causative agent was identified in 28.1% of cases. Among the viral etiologies, influenza virus was detected by PCR in 28.9% of patients, SARS-CoV-2 in 23.15%, and other viral agents were identified in four cases.

The mean duration of hospital stay for ARDS patients was 19.4 ± 13.8 days, while the mean duration of ICU stay was 11.3 ± 9.04 days.

The mean duration on ventilatory support was 9.2 ± 7.4 days. The mean number of ventilator-free days (VFDs) was 10.0 ± 10.8 days. Mechanical ventilation parameters and values of arterial blood gas are shown in Table 2.

The most prevalent comorbidities among the analyzed ARDS patients were arterial hypertension (43.8%), diabetes mellitus (24.0%), chronic obstructive pulmonary disease (17.4%), and ischemic cardiomyopathy (11.6%).

The 28-day mortality rate was 46.3%, with cumulative survival rates of 86% at day 7, 72.7% at day 14, 60.3% at day 21, and 53.7% at day 28 (Figure 1).

Older age was significantly associated with an increase in 28-day mortality rates (*p* = 0.048), and patients with higher Charlson comorbidity index scores experienced increased mortality (*p* = 0.006).

Disease severity, based on the Berlin classification, also correlated with mortality, with a 56.9% mortality rate in severe ARDS cases compared to 36.5% in mild to moderate ARDS cases (*p* = 0.025).

Diabetes mellitus had the most significant impact on 28-day mortality (*p* = 0.051), with a mortality rate of 62.1% in diabetic patients compared to 41.3% in non-diabetic patients. The presence of malignancy was associated with a higher 28-day mortality rate of 72.7%, in contrast to 43.6% in patients without malignancy (*p* = 0.110). Other comorbidities did not show a statistically significant correlation with 28-day mortality in ARDS patients.

Among etiological factors, a proven SARS-CoV-2 infection from a nasopharyngeal swab had the most pronounced impact on 28-day mortality (*p* = 0.009).

### sRAGE and 28-Day Mortality

Since the serum biomarker of alveolocapillary membrane damage in this study was found to be non-normally distributed, comparisons were performed using the Mann–Whitney U test, with results presented as Box–Whisker plots.

The range of sRAGE biomarker values spanned from 468 to 30,000 pg/mL, with a median value of 5229 pg/mL (IQR 10,285). Patients who did not survive had higher median sRAGE levels compared to those who survived at day 28 (5852 pg/mL vs. 4479 pg/mL, respectively), with a *p*-value of 0.084 (Table 3 and Figure 2). 

Table 4 and Figure 3 report the values for the area under the ROC curve (ROC area), optimal cut-off values, and corresponding sensitivity and specificity for sRAGE. Additionally, the table also reports the calculated positive predictive value (PPV) and negative predictive value (NPV) for this biomarker.

Figure 4 presents the correlation between ICU LOS, VFD, and the cut-off values of sRAGE using density plots, providing a depiction of the distribution patterns across survival and non-survival groups.

For ICU LOS, the distributions differ between 28-day survivors (Panel A) and non-survivors (Panel B). Among survivors, patients with higher sRAGE levels (≥16,500 pg/mL) demonstrate a shift toward longer ICU stays compared to those with lower sRAGE levels (<16,500 pg/mL). In contrast, non-survivors exhibit a more concentrated distribution of ICU LOS, with higher sRAGE levels showing a modest tendency toward prolonged ICU stays. These findings suggest a potential association between elevated sRAGE levels and extended ICU care, particularly in survivors.

For VFD, the distributions (Panels C and D) reveal a negative relationship between sRAGE levels and ventilator-free days. Among 28-day survivors (Panel C), higher sRAGE levels (≥16,500 pg/mL) are associated with fewer ventilator-free days, indicating a greater reliance on mechanical ventilation. In non-survivors (Panel D), the VFD distributions are heavily skewed toward zero for both sRAGE categories, reflecting the absence of ventilator-free days in this group.

These findings collectively suggest that elevated sRAGE levels are associated with poorer clinical outcomes, including prolonged ICU stays and reduced ventilator-free days, particularly among survivors. Further research is warranted to elucidate the mechanisms underlying these associations.

## 4. Discussion

In our study, the mean sRAGE concentration was 5229 pg/mL. Although higher sRAGE levels were observed in patients who did not survive, no statistically significant difference was found between survivors and non-survivors. The cut-off value for predicting 28-day mortality was determined to be >16,500 pg/mL, with a sensitivity of 30.4% and specificity of 86.9%.

One of the earliest studies on the association between sRAGE and 28-day mortality in ARDS was conducted in 2008 by Calfee et al., identifying sRAGE as a potential independent predictor of 28-day mortality (OR for death 1.38 (95% CI 1.13 to 1.68) per 1 log increment in RAGE; *p* = 0.002). This pioneering study concluded that elevated plasma sRAGE levels were associated with increased mortality and poor oxygenation, highlighting sRAGE’s role as a biomarker of lung epithelial injury [16]. Both studies highlighted the prognostic potential of sRAGE. Our study demonstrated a trend toward higher sRAGE levels in non-survivors, but this finding did not reach statistical significance. In contrast to our study, Calfee et al. measured sRAGE at two time points: baseline and Day 3. Their dynamic measurements offered valuable insights into the potential role of sRAGE in disease progression and highlighted its utility in risk stratification. However, our study, being non-interventional, did not include repeated measurements, which represents a notable limitation when comparing findings.

Jabadon et al. conducted a meta-analysis exploring baseline plasma sRAGE levels in association with 90-day mortality in ARDS, aiming to determine the prognostic potential of sRAGE for assessing the severity of epithelial injury and predicting clinical outcomes. Their results indicated that sRAGE levels were significantly higher among non-survivors (4335 pg/mL vs. 3198 pg/mL, *p* = 0.002) suggesting that sRAGE could serve as an independent predictor of mortality. Although cut-off values were not defined in their study, consistently elevated sRAGE levels were linked to worse outcomes, especially in patients with severe ARDS, often due to mechanical ventilation-related lung injury. The statistical significance of this study is underscored by the finding that log-transformed sRAGE values showed a clear correlation with mortality rates, mechanical ventilation parameters, and baseline patient characteristics [14]. Our findings also demonstrated elevated sRAGE levels in non-survivors. However, unlike the meta-analysis, our study included a defined cut-off, which showed moderate specificity but poor sensitivity. The observed standard deviation (9833) was higher than anticipated, based on data published by Jabaudon et al., resulting in a reduction in a study’s statistical power, potentially limiting the ability to detect significant differences as initially planned. A consistent observation across both studies is the association between higher sRAGE levels and ARDS severity.

In another study by Jabadon et al., it was revealed that patients with ARDS had elevated arterial, central venous, and alveolar levels of sRAGE and its isoforms compared to control groups (baseline 11 ng/mL vs. 104 ng/mL, *p* = 0.05, 14 ng/mL vs. 104 ng/mL, *p* = 0.06, 47 ng/mL vs. 103 ng/mL, *p* = 0.4, for arterial, central venous and alveolar levels, respectively, AUC 0.79). These elevated levels were associated with ARDS severity, as measured by the PaO_2_/FiO_2_ ratio, marking the first study to employ repeated measurements of key RAGE isoforms and ligands in ARDS [17]. Our study also observed elevated sRAGE levels correlating with ARDS severity but did not include detailed compartmental analyses or repeated measurements. In contrast, Jabaudon et al. emphasized isoforms and conducted repeated measurements, whereas our study focused primarily on serum sRAGE levels for prognostication.

Peukert et al. examined the role of increased alveolar epithelial damage markers in pulmonary superinfection in ARDS. They found that the serum concentrations of SP-D and sRAGE were both significantly increased in ARDS patients with pulmonary superinfections compared to ARDS patients who did not develop pulmonary superinfections (*p* = 0.0397 and *p* = 0.049, respectively), emphasizing that tracking dynamic biomarker changes over time offers improved prognostic insights, particularly in cases complicated by pulmonary superinfections [18]. Our study did not examine pulmonary superinfections or sRAGE dynamics over time.

More recently, research by Yehya et al. on pediatric ARDS investigated sRAGE dynamics along with other inflammation and tissue injury markers. Their findings showed significantly higher sRAGE concentrations in patients with pulmonary ARDS compared to those with indirect ARDS, suggesting that sRAGE may serve as a phenotypic marker distinguishing ARDS subtype, which is critical for tailoring therapeutic approaches to the underlying pathophysiological mechanisms [19]. Our study did not distinguish between ARDS subtypes but aligned with the general observation of elevated sRAGE levels in contexts of more severe or direct epithelial injury.

Research during the COVID-19 pandemic further investigated the role of sRAGE in ARDS. In our study, 23% of the patients had COVID-ARDS. A study by Lim et al. proposed a threshold of 25,000 pg/mL for high-risk patients, suggesting that this biomarker could be instrumental in predicting disease progression and guiding ventilatory strategies [20]. The discrepancy in cut-off values observed in our results may be attributed to the greater heterogeneity of the patient population included in our study. This heterogeneity highlights the need for future research utilizing more homogeneous patient cohorts to ensure more precise and clinically relevant findings. Both studies concur on sRAGE as a prognostic marker for mortality, although our study did not investigate its role in disease progression.

Studies from this period showed that elevated sRAGE levels in COVID-19 patients were linked to increased mortality risk, particularly among those on mechanical ventilation and corticosteroid therapy. A study by Butcher et al. specifically examined patients treated with dexamethasone, where high sRAGE levels were associated with a significantly increased mortality risk (2852 pg/mL vs. 1014 pg/mL, *p* < 0.001), underscoring the potential role of sRAGE for stratifying COVID-19 patients by risk, especially those receiving corticosteroid treatment [21]. Although our study observed elevated sRAGE levels in non-survivors, the mean levels were considerably higher in their cohort. This discrepancy is likely attributable to differences in patient populations. Both studies emphasize the prognostic significance of sRAGE while highlighting variability in cut-off levels across cohorts.

A primary limitation of this study is its single-center design, which may restrict the generalizability of the findings to broader populations. Additionally, potential confounders, such as unmeasured genetic variability in sRAGE levels, could influence the results and should be considered when interpreting the study’s outcomes.

The limitations of sRAGE as a prognostic biomarker for ARDS stem from its low sensitivity of 30.4%, which indicates a significant proportion of patients with poor outcomes may not be identified by this marker alone. This reduces its utility for early, reliable detection of high-risk patients. While the specificity of 86.9% suggests sRAGE can effectively rule out patients unlikely to have poor outcomes, its limited sensitivity undermines its standalone prognostic value. These performance metrics highlight the need for sRAGE to be used in conjunction with other biomarkers or clinical indicators to improve overall predictive accuracy and enhance its applicability in clinical decision-making for ARDS management.

## 5. Conclusions

Based on the findings of our study and corroborating evidence from previous research, while sRAGE demonstrates promising trends as a prognostic biomarker for ARDS, its predictive performance in this study was limited. Validation in larger, multicenter studies is required to confirm its clinical applicability. However, to enhance predictive precision and account for the multifaceted, heterogeneous nature of ARDS, it is imperative to implement an integrative approach that combines sRAGE with a panel of other biomarkers. This multi-biomarker strategy would refine risk stratification, yielding a more nuanced assessment and offering deeper insights into the complex pathophysiological mechanisms that contribute to mortality in ARDS. A larger, multicenter study is warranted to validate these findings, and the integration of multiple biomarkers may enhance prognostic accuracy and improve clinical outcomes.

## Figures and Tables

**Figure 1 medicina-61-00229-f001:**
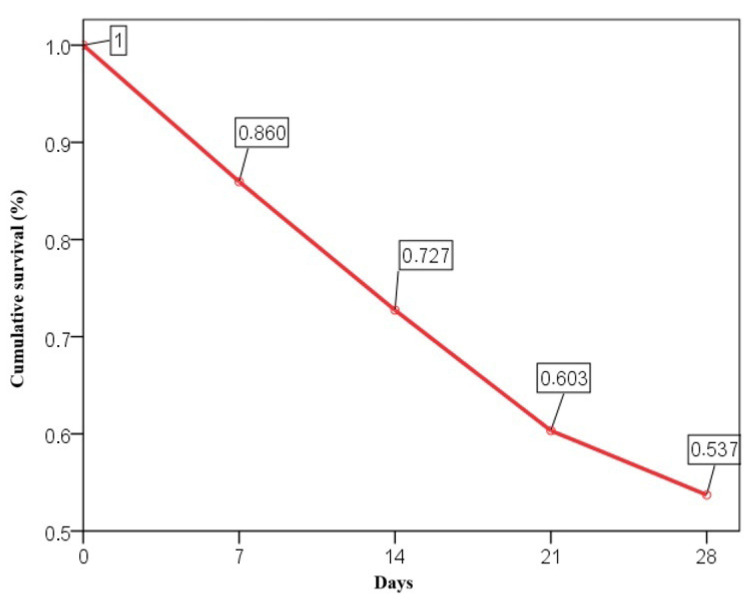
Cumulative survival in ARDS patients.

**Figure 2 medicina-61-00229-f002:**
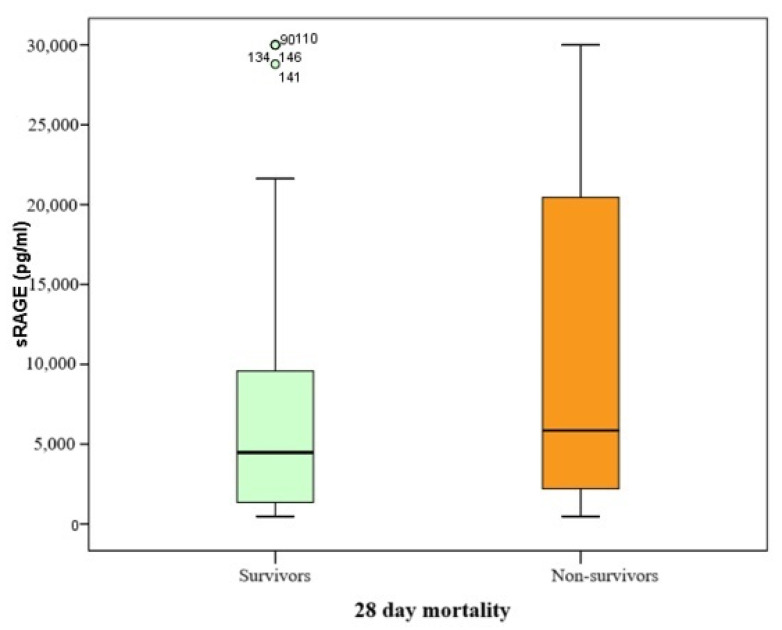
Box–Whisker plot of sRAGE and mortality at day 28.

**Figure 3 medicina-61-00229-f003:**
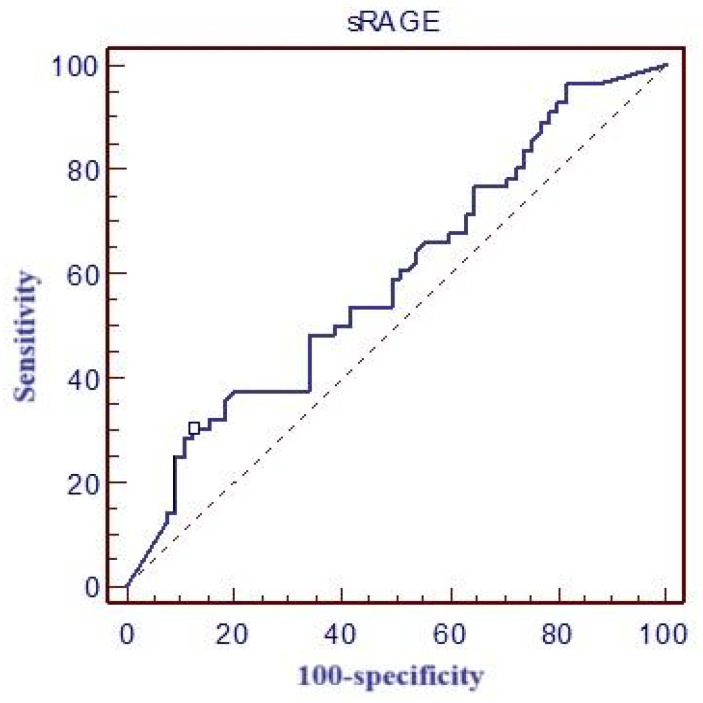
ROC curve of sRAGE and mortality at day 28.

**Figure 4 medicina-61-00229-f004:**
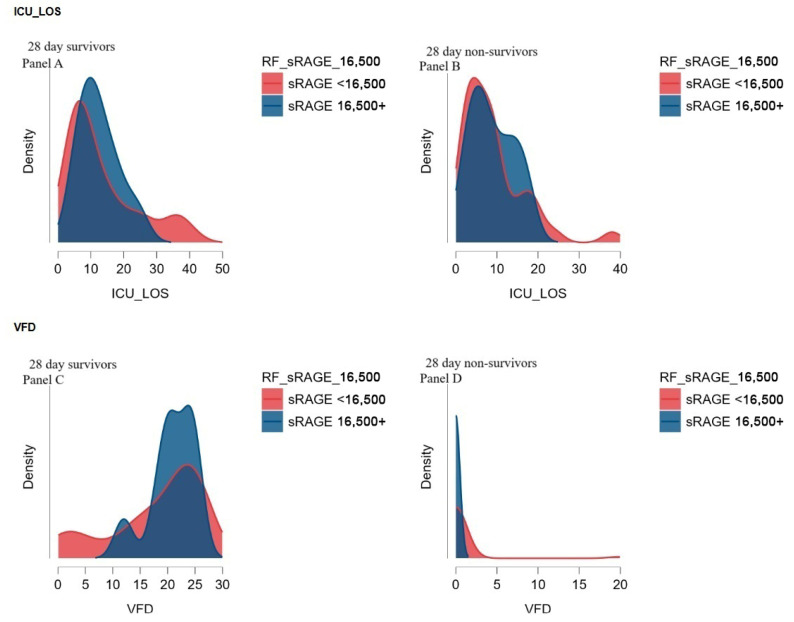
(**A**–**D**) The distribution of sRAGE values in relation to mortality outcomes. Panels (**A**,**B**) show the density distribution of ICU length of stay for 28-day survivors and non-survivors, with higher sRAGE levels (blue) correlating with longer ICU stays in survivors. Panels (**C**,**D**) depict the ventilator-free day distributions, revealing that higher sRAGE levels are associated with fewer ventilator-free days, particularly in non-survivors (**D**). The *y*-axis in each panel represents the density of the distribution, indicating the relative frequency of the corresponding *x*-axis values.

**Table 1 medicina-61-00229-t001:** Description of the cohort.

Variables	All Patients *n* = 121	Survivors at Day 28 *n* = 65	Non-Survivors at Day 28 *n* = 56	*p* Value
Age (years)	55.5	53.0	58.4	0.048
Male (%)	63.6%	53.2%	48.5%	
Severity of illness				
APACHE II, (mean SD)	17.6 (6.53)	15.6 (5.44)	19.9 (6.98)	0.002
SOFA score, mean (SD)	7.26 (3.26)	6.4 (2.4)	8.27 (3.83)	0.007
SAPS 2 score, mean (SD)	43.8 (16.7)	38.6 (13.0)	49.8 (18.5)	0.001
Charlson comorbidity index mean (SD)	2.38 (1.93)	1.91 (1.91)	2.93 (1.82)	0.002
C-reactive protein (mg/L) mean (SD)	191 (113)	200 (118)	179 (107)	0.267
Procalcitonin (ng/mL) mean (SD)	9.29 (22.4)	8.61 (16.8)	10.1 (27.8)	0.726
Lactate (mmol/L) mean (SD)	2.17 (1.77)	2.3 (1.73)	2.71 (1.8)	0.161

**Table 2 medicina-61-00229-t002:** Average values of mechanical ventilation parameters and arterial blood gas parameters.

Parameter	Median (IQR)
Tidal volume (mL)	400 (70)
PEEP (cmH_2_O)	10 (2)
FiO_2_	0.7 (0.33)
Plateau pressure (cmH_2_O)	24 (6)
Driving pressure (cmH_2_O)	12 (5)
Static compliance (mL/cmH_2_O)	33.3 (13.9)
PaO_2_/FiO_2_	106 (84)
pCO_2_ (kPa)	6.1 (2.5)
pH	7.28 (0.13)

**Table 3 medicina-61-00229-t003:** sRAGE and mortality on day 28.

sRAGE (pg/mL)	n	Mean (SD)	Min–Max	Median (IQR)	*p*
Survivors	65	7478 (8603)	468–30,000	4479 (9094)	0.084
Non-survivors	56	10,974 (5852)	468–30,000	5852 (19,135)	
Total	121	9096 (9833)	468–30,000	5229 (10,285)	

**Table 4 medicina-61-00229-t004:** ROC analysis of mortality at day 28.

	ROC Area	*p* Value	Cut-Off	Sensitivity	Specificity	PPV	NPV
**sRAGE (pg/mL)**	0.591	0.079	>16,500	30.4%	86.9%	65.4	58.9

## Data Availability

The data presented in this study are available on request from the corresponding author due to privacy reasons.

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
