# Peer review of "sRAGE as a Prognostic Biomarker in ARDS: Insights from a Clinical Cohort Study"

_medicina, 2025, doi:10.3390/medicina61020229_

Round 1
Reviewer 1 Report
Comments and Suggestions for Authors
This study clearly outlines the severity and high mortality rate of ARDS and highlights the current lack of definitive biomarkers for diagnosis and prognostic stratification. This study design is a retrospective cohort study with data collected from a tertiary care ICU in Serbia from January 2021 to June 2023. It includes detailed inclusion and exclusion criteria to ensure data accuracy and consistency. This study collects a wide range of data, including demographics, comorbidities, clinical severity scores, laboratory markers, and ventilatory parameters. This study uses various statistical methods (e.g., Chi-square test, Fisher's exact test, t-test, Mann-Whitney U test, logistic regression models, and Kaplan-Meier survival curves) to evaluate the prognostic value of sRAGE for 28-day mortality. This study received ethical committee approval and adhered to relevant data protection laws and institutional policies.
Although non-survivors had higher median sRAGE levels than survivors, the difference did not reach statistical significance (p=0.084). This study did not account for potential confounding factors, such as unmeasured genetic variability that could affect sRAGE levels. This study recommends conducting multicenter studies to validate these findings and integrating multiple biomarkers to improve predictive accuracy.
Overall, the study provides valuable insights into the potential of sRAGE as a prognostic biomarker for ARDS, but further multicenter studies are needed to validate the results and improve predictive accuracy.
Author Response
Comment 1: This study clearly outlines the severity and high mortality rate of ARDS and highlights the current lack of definitive biomarkers for diagnosis and prognostic stratification. This study design is a retrospective cohort study with data collected from a tertiary care ICU in Serbia from January 2021 to June 2023. It includes detailed inclusion and exclusion criteria to ensure data accuracy and consistency. This study collects a wide range of data, including demographics, comorbidities, clinical severity scores, laboratory markers, and ventilatory parameters. This study uses various statistical methods (e.g., Chi-square test, Fisher's exact test, t-test, Mann-Whitney U test, logistic regression models, and Kaplan-Meier survival curves) to evaluate the prognostic value of sRAGE for 28-day mortality. This study received ethical committee approval and adhered to relevant data protection laws and institutional policies.
Although non-survivors had higher median sRAGE levels than survivors, the difference did not reach statistical significance (p=0.084). This study did not account for potential confounding factors, such as unmeasured genetic variability that could affect sRAGE levels. This study recommends conducting multicenter studies to validate these findings and integrating multiple biomarkers to improve predictive accuracy.
Overall, the study provides valuable insights into the potential of sRAGE as a prognostic biomarker for ARDS, but further multicenter studies are needed to validate the results and improve predictive accuracy.
Response 1: Thank you for your thorough and constructive feedback on our study. We greatly appreciate your detailed evaluation and insights. Below, we address the key points raised:
- Study Design and Data Collection:
We are glad you found the inclusion and exclusion criteria, as well as the data collected, to be detailed and robust. Ensuring data accuracy and consistency was a primary objective during our study design. - Statistical Methods:
We appreciate your recognition of the comprehensive statistical analyses employed. These methods were selected to ensure robust evaluation of the prognostic value of sRAGE for 28-day mortality. - Significance of sRAGE Levels:
Regarding the finding that non-survivors had higher median sRAGE levels without reaching statistical significance (p=0.084), we acknowledge this limitation. As noted, the small sample size could have reduced the power to detect a statistically significant difference. We have emphasized this limitation in the Discussion section and proposed larger, multicenter studies as a follow-up. - Confounding Factors:
Your observation about unmeasured genetic variability potentially influencing sRAGE levels is valid. While this was beyond the scope of our current study, we agree it is an important consideration for future research. We have now highlighted this point in the Discussion and added it to the list of study limitations. - Recommendation for Multicenter Studies and Biomarker Integration:
We are grateful for your support of our recommendation for multicenter studies and biomarker integration. We have expanded on this point to underline its importance in advancing the field.
We hope these revisions address your comments and enhance the clarity and impact of the manuscript. Thank you once again for your thoughtful review.
Reviewer 2 Report
Comments and Suggestions for Authors
The manuscript addresses a relevant topic in the context of ARDS, focusing on the biomarker sRAGE. While the subject is of interest, the study presents methodological and interpretive limitations that need to be addressed to strengthen the validity of the work and its clinical applicability.
2. Abstract
Strengths:
The abstract clearly communicates the purpose of the study and its methodology.
It highlights the potential use of sRAGE as a prognostic biomarker.
Limitations:
The main findings (p=0.084) are not statistically significant, which should be explicitly mentioned to avoid misinterpretations.
The limitations of the retrospective design are not adequately emphasized.
Recommendation: Explicitly mention the lack of statistical significance in the main findings and better contextualize the clinical applicability of the biomarker.
3. Introduction
Strengths:
Provides adequate context regarding the importance of biomarkers in ARDS.
Includes relevant background on sRAGE and its relationship with pulmonary pathophysiology.
Limitations:
Key studies exploring sRAGE in larger cohorts, such as Calfee et al. (2008) and Jabaudon et al. (2018), are not cited.
Comparisons with other biomarkers used in ARDS, such as SP-D or inflammatory markers (Blondonnet et al., 2016), are missing.
Recommendation: Include references like Jabaudon et al. (2018), which demonstrated the utility of sRAGE as an independent predictor of mortality in ARDS, and Blondonnet et al. (2016), which discusses complementary biomarkers.
4. Methodology
Strengths:
The retrospective design, inclusion/exclusion criteria, and ELISA technique for sRAGE measurement are well described.
Appropriate statistical tools, such as logistic regression and Kaplan-Meier curves, are employed.
Limitations:
The sample size is limited, reducing the power to detect statistically significant differences, as evidenced by the p=0.084 result.
Dynamic biomarker data are not included, a critical aspect explored by Yehya et al. (2024) and Peukert et al. (2023).
Recommendation: Justify the sample size more clearly and consider how the lack of longitudinal biomarker data affects the conclusions. Reference studies like Yehya et al. (2024) to highlight the importance of biomarker dynamics.
5. Results
Strengths:
Demographic and clinical data are well presented in tables and graphs.
The correlation between elevated sRAGE levels and higher mortality is adequately described.
Limitations:
Secondary outcomes (e.g., ventilator-free days) are not thoroughly analyzed, limiting the interpretation of sRAGE’s clinical impact.
The relatively low predictive value of the biomarker is not discussed in depth compared to other studies (Calfee et al., 2008; Jabaudon et al., 2018).
Recommendation: Compare the findings with previous studies, such as the sRAGE cutoffs in ARDS proposed by Lim et al. (2021) (>25,000 pg/mL), and discuss the implications of the observed differences.
6. Discussion
Strengths:
The findings are contextualized in relation to previous studies, highlighting similar trends.
The limitations of the single-center design are acknowledged.
Limitations:
The low sensitivity (30.4%) of the identified cutoff (>16,500 pg/mL) is insufficiently addressed.
The potential combination of sRAGE with other biomarkers is not explored, as suggested by Blondonnet et al. (2016) and Spadaro et al. (2019).
Recommendation: Expand the discussion on the clinical utility of sRAGE in combination with inflammatory biomarkers. Consider including studies like Butcher et al. (2022), which explore sRAGE in the context of COVID-19-ARDS.
7. Conclusions
Strengths:
The conclusions reflect the findings presented.
The need for multicenter studies to validate sRAGE’s utility is emphasized.
Limitations:
The claim of “substantial potential” for sRAGE appears overstated given the low predictive performance reported.
Recommendation: Reformulate the conclusions to acknowledge that sRAGE shows promising trends but requires further validation in broader, multicenter studies for clinical implementation.
8. References
Strengths:
Relevant citations supporting ARDS pathophysiology and sRAGE as a biomarker are included.
Limitations:
Key studies are missing, such as:
Calfee et al. (2008), on the association between sRAGE and mortality in ARDS.
Jabaudon et al. (2018), which establishes sRAGE as an independent biomarker in multicenter studies.
Lim et al. (2021), on sRAGE in COVID-19-ARDS.
Recommendation: Expand the bibliography to strengthen the theoretical framework and contextualize the results within the broader biomarker research landscape in ARDS.
Author Response
Comment 1:
- Abstract
Strengths: The abstract clearly communicates the purpose of the study and its methodology.
It highlights the potential use of sRAGE as a prognostic biomarker.
Limitations: The main findings (p=0.084) are not statistically significant, which should be explicitly mentioned to avoid misinterpretations.
The limitations of the retrospective design are not adequately emphasized.
Recommendation: Explicitly mention the lack of statistical significance in the main findings and better contextualize the clinical applicability of the biomarker.
Response 1:
Thank you for your feedback. We agree that explicitly mentioning the lack of statistical significance (p=0.084) in the main findings is essential to avoid misinterpretation. Additionally, we will emphasize the limitations of the retrospective design in the abstract.
Corrected text: " The main findings, while indicative of a trend toward higher sRAGE levels in non-survivors, did not reach statistical significance (p=0.084). sRAGE demonstrates potential as a prognostic biomarker in ARDS and has moderate correlation with 28-day mortality. Integrating sRAGE with other biomarkers could enhance risk stratification and guide therapeutic decisions. The retrospective design limits the ability to establish causation, underscoring the need for multicenter prospective studies."
Comment 2:
3. Introduction
Strengths: Provides adequate context regarding the importance of biomarkers in ARDS.
Includes relevant background on sRAGE and its relationship with pulmonary pathophysiology.
Limitations: Key studies exploring sRAGE in larger cohorts, such as Calfee et al. (2008) and Jabaudon et al. (2018), are not cited. Comparisons with other biomarkers used in ARDS, such as SP-D or inflammatory markers (Blondonnet et al., 2016), are missing.
Recommendation: Include references like Jabaudon et al. (2018), which demonstrated the utility of sRAGE as an independent predictor of mortality in ARDS, and Blondonnet et al. (2016), which discusses complementary biomarkers.
Response 2:
We appreciate the recommendation to include key studies, such as Jabaudon et al. (2018) and Blondonnet et al. (2016), as well as comparisons with complementary biomarkers like SP-D. These references will be integrated to provide a comprehensive background. Also, those references are also pointed out in Disscussion.
Added text: " Jabaudon et al. conducted a comprehensive analysis demonstrating that elevated baseline plasma sRAGE levels are independently associated with increased 90-day mortality in ARDS patients. Their findings highlighted the utility of sRAGE as an indicator of disease severity and a predictor of clinical outcomes, reinforcing its potential as a key component in risk stratification and management protocols for ARDS. Notably, their work emphasized the pathophysiological link between alveolar epithelial damage and the systemic inflammatory response, which contributes to the progression of ARDS. Complementing this, Blondonnet et al. explored the pathophysiological significance of sRAGE and its related ligands in ARDS, identifying it as a valuable marker for assessing lung injury severity. Their study underscored the biomarker's dual role in diagnosing the extent of epithelial damage and in guiding therapeutic interventions, particularly in heterogenous clinical settings. They also compared sRAGE with other biomarkers, such as surfactant protein-D (SP-D), another marker of epithelial injury, noting that while SP-D is indicative of type II alveolar cell damage, sRAGE uniquely reflects type I alveolar epithelial injury, offering complementary insights into the mechanisms underlying ARDS. The study proposed that sRAGE could serve not only as a prognostic tool but also as a means to tailor individualized treatment strategies, aligning with the broader goal of personalized medicine in critical care. These findings collectively support the hypothesis that sRAGE may provide critical insights into ARDS pathophysiology and prognosis."
Comment 3:
4. Methodology
Strengths: The retrospective design, inclusion/exclusion criteria, and ELISA technique for sRAGE measurement are well described. Appropriate statistical tools, such as logistic regression and Kaplan-Meier curves, are employed.
Limitations: The sample size is limited, reducing the power to detect statistically significant differences, as evidenced by the p=0.084 result. Dynamic biomarker data are not included, a critical aspect explored by Yehya et al. (2024) and Peukert et al. (2023).
Recommendation: Justify the sample size more and consider how the lack of longitudinal biomarker data affects the conclusions. Reference studies like Yehya et al. (2024) to highlight the importance of biomarker dynamics.
Response 3:
Thank you for pointing out the limitations in sample size and lack of dynamic biomarker data. We will provide a clearer justification for the sample size and acknowledge the implications of not including longitudinal data, referencing Yehya et al. (2024) and Peukert et al. (2023) in Discussion part.
Added text: " The sample size was based on assumed in-hospital rate of death of 50% and expected sRAGE standard deviation of 4000. Thus an enrollment of 126 would have a power of 80% (at a two-sided alpha level of 0.05) to detect difference in sRAGE means of 2000 between groups. While limited, the sample size reflects the constraints of a single-center retrospective study. We acknowledge that incorporating dynamic biomarker data could provide deeper insights into the prognostic role of sRAGE. Additionally, the lack of investigation into sRAGE's performance in predicting disease progression represents a limitation of our study."
Also, in the Discussion part we will add "The observed standard deviation (9833) was higher than anticipated, based on data published by Jabaudon et al, resulting in a reduction in a study’s statistical power, potentially limiting the ability to detect significant difference as initially planned. "
Comment 4:
5. Results
Strengths:
Demographic and clinical data are well presented in tables and graphs.
The correlation between elevated sRAGE levels and higher mortality is adequately described.
Limitations:
Secondary outcomes (e.g., ventilator-free days) are not thoroughly analyzed, limiting the interpretation of sRAGE’s clinical impact.
The relatively low predictive value of the biomarker is not discussed in depth compared to other studies (Calfee et al., 2008; Jabaudon et al., 2018).
Recommendation: Compare the findings with previous studies, such as the sRAGE cutoffs in ARDS proposed by Lim et al. (2021) (>25,000 pg/mL), and discuss the implications of the observed differences.
Response 4:
We appreciate the suggestion to analyze secondary outcomes more thoroughly and compare findings with other studies. We will expand on the analysis of ventilator-free days and ICU length of stay.
Added text: "Graph3. provides insights into the relationship between sRAGE levels and secondary clinical outcomes. For ICU LOS, the distributions differ between 28-day survivors (Panel A) and non-survivors (Panel B). Among survivors, patients with higher sRAGE levels (≥16,500pg/ml) demonstrate a shift toward longer ICU stays compared to those with lower sRAGE levels (<16,500pg/ml). In contrast, non-survivors exhibit a more concentrated distribution of ICU LOS, with higher sRAGE levels showing a modest tendency toward prolonged ICU stays. These findings suggest a potential association between elevated sRAGE levels and extended ICU care, particularly in survivors.
For VFD, the distributions (Panels C and D) reveal a negative relationship between sRAGE levels and ventilator-free days. Among 28-day survivors (Panel C), higher sRAGE levels (≥16,500pg/ml) are associated with fewer ventilator-free days, indicating a greater reliance on mechanical ventilation. In non-survivors (Panel D), the VFD distributions are heavily skewed toward zero for both sRAGE categories, reflecting the absence of ventilator-free days in this group.
These findings collectively suggest that elevated sRAGE levels are associated with poorer clinical outcomes, including prolonged ICU stays and reduced ventilator-free days, particularly among survivors. Further research is warranted to elucidate the mechanisms underlying these associations."
Thank you for your valuable feedback and for highlighting the importance of including discussions on Lim et al (2021). We appreciate your suggestion, and we would like to note that these studies have already been discussed in the Discussion section of the manuscript. Specifically: Research during the COVID-19 pandemic further investigated the role of sRAGE in ARDS. In our study, 23% of the patients had Covid-ARDS. Study by Lim et al. proposed a threshold of 25,000 pg/mL for high-risk patients, suggesting that this biomarker could be instrumental in predicting disease progression and guiding ventilatory strategies [20]. Our cutoff was notably lower than the one proposed by Lim et al., potentially reflecting differences in study populations. Both studies concur on sRAGE as a prognostic marker for mortality, although our study did not investigate its role in disease progression.
Also, by pointing out observed differences between our papers, we will add "The discrepancy in cutoff values observed in our results may be attributed to the greater heterogeneity of the patient population included in our study. This heterogeneity highlights the need for future research utilizing more homogeneous patient cohorts to ensure more precise and clinically relevant findings".
Comment 5:
6. Discussion
Strengths:
The findings are contextualized in relation to previous studies, highlighting similar trends.
The limitations of the single-center design are acknowledged.
Limitations:
The low sensitivity (30.4%) of the identified cutoff (>16,500 pg/mL) is insufficiently addressed.
The potential combination of sRAGE with other biomarkers is not explored, as suggested by Blondonnet et al. (2016) and Spadaro et al. (2019).
Recommendation: Expand the discussion on the clinical utility of sRAGE in combination with inflammatory biomarkers. Consider including studies like Butcher et al. (2022), which explore sRAGE in the context of COVID-19-ARDS.
Response 5:
We agree that the low sensitivity of the identified cutoff should be addressed and that sRAGE’s clinical utility may improve when combined with other biomarkers.
Added text: "The limitations of sRAGE as a prognostic biomarker for ARDS stem from its low sensitivity of 30.4%, which indicates a significant proportion of patients with poor outcomes may not be identified by this marker alone. This reduces its utility for early, reliable detection of high-risk patients. While the specificity of 86.9% suggests sRAGE can effectively rule out patients unlikely to have poor outcomes, its limited sensitivity undermines its standalone prognostic value. These performance metrics highlight the need for sRAGE to be used in conjunction with other biomarkers or clinical indicators to improve overall predictive accuracy and enhance its applicability in clinical decision-making for ARDS management. "
Thank you for your insightful suggestion regarding the inclusion of Butcher et al. in our manuscript. We appreciate its relevance to the subject matter. We would like to kindly point out that this paper has already been discussed in the Discussion section where we have compared its findings with our results. However, if there are additional aspects or specific points you feel should be further elaborated upon, we are more than willing to revise the text accordingly.
Comment 6:
7. Conclusions
Strengths: The conclusions reflect the findings presented. The need for multicenter studies to validate sRAGE’s utility is emphasized.
Limitations: The claim of “substantial potential” for sRAGE appears overstated given the low predictive performance reported.
Recommendation: Reformulate the conclusions to acknowledge that sRAGE shows promising trends but requires further validation in broader, multicenter studies for clinical implementation.
Response 6:
We appreciate the recommendation to temper the conclusions. The revised conclusion will reflect the promising trends observed while emphasizing the need for further validation.
Corrected text: "While sRAGE demonstrates promising trends as a prognostic biomarker for ARDS, its predictive performance in this study was limited. Validation in larger, multicenter studies is required to confirm its clinical applicability."
Comment 7:
8. References
Strengths:
Relevant citations supporting ARDS pathophysiology and sRAGE as a biomarker are included.
Limitations:
Key studies are missing, such as:
Calfee et al. (2008), on the association between sRAGE and mortality in ARDS.
Jabaudon et al. (2018), which establishes sRAGE as an independent biomarker in multicenter studies.
Lim et al. (2021), on sRAGE in COVID-19-ARDS.
Recommendation: Expand the bibliography to strengthen the theoretical framework and contextualize the results within the broader biomarker research landscape in ARDS.
Response 7:
Thank you for highlighting key references. I would like to note that the mentioned papers are already included in the reference list of this manuscript.
- Jabaudon M, Blondonnet R, Pereira B, Cartin-Ceba R, Lichtenstern C, Mauri T, et al. Plasma sRAGE is independently associated with increased mortality in ARDS: a meta-analysis of individual patient data. Intensive Care Med. 2018.
- Calfee CS, Ware LB, Eisner MD, Parsons PE, Thompson BT, Wickersham N, et al. Plasma receptor for advanced glycation end products and clinical outcomes in acute lung injury. Thorax. 2008 May 20;63(12):1083–9.
- Lim A, Radujkovic A, Weigand MA, Merle U. Soluble receptor for advanced glycation end products (sRAGE) as a biomarker of COVID-19 disease severity and indicator of the need for mechanical ventilation, ARDS and mortality. Ann Intensive Care. 2021 Dec 22;11(1):50.
Reviewer 3 Report
Comments and Suggestions for Authors
The paper is quite well written. The article covers a very interesting and current topic. I have some comments:
1) Abstract. Results: A cohort of 121 patients (mean age 55.5 years; 63.6% male) was analyzed. Non-survivors exhibited higher median sRAGE levels than survivors (5,852 vs. 4,479 pg/mL), though statistical significance was not reached (p=0.084). The optimal sRAGE cutoff for predicting mortality was >16,500 pg/mL (sensitivity 30.4%, speci-ficity 86.9%). Higher sRAGE levels correlated with increased disease severity and mortality risk. I suggest to underline the most important data to support the conclusions.
2) Abstract. Conclusions: sRAGE demonstrates potential as a prognostic biomarker in ARDS and has moderate correlation with 28-day mortality. Integrating sRAGE with other biomarkers could enhance risk stratification and guide therapeutic decisions. Further multicenter studies are warranted to vali-date these findings. Abstract might be beneficial to include a sentence that briefly summarizes the key findings of the study. This can provide readers with a quick overview of the research.
3) Identifying a serum biomarker that correlates with clinical presentation and labora-91 tory findings could significantly aid in disease diagnosis and enable clinicians to stratify 92 ARDS severity more accurately, as well as assisting in sequencing interventions. Incorpo-93 rating the timing and sequencing of interventions into ARDS management protocols is 94 crucial to standardizing care, improving outcome consistency, and effectively stratifying 95 disease severity [14]. I suggest to underline the aim of the study and the novelty of the study.
4) 2.4. Statistical Analysis 148 Summaries of patient characteristics, sRAGE levels, and clinical outcomes were re-149 ported as means with standard deviations or medians with interquartile ranges, 150.. I suggest to add the statistically significant p-value.
5) 3. Results. I suggest to underline the most important statistically significant values to support the conclusions.
6) 4. Discussion 276 In our study, the mean sRAGE concentration was 5,229 pg/mL. Although higher 277 sRAGE levels were observed in patients who did not survive, no statistically significant 278 difference was found between survivors and non-survivors. The cut-off value for predict- 279 ing 28-day mortality was determined to be >16,500 pg/mL, with a sensitivity of 30.4% and 280 specificity of 86.9%.... The discussion section needs to be improved. It is necessary to be more concise in the presentation of the facts, clarifying the results obtained and comparing them with previous or similar studies. However, it is interesting to answer the questions that arise from these results, backed up by published literature.
Author Response
Comment1:
Abstract. Results: A cohort of 121 patients (mean age 55.5 years; 63.6% male) was analyzed. Non-survivors exhibited higher median sRAGE levels than survivors (5,852 vs. 4,479 pg/mL), though statistical significance was not reached (p=0.084). The optimal sRAGE cutoff for predicting mortality was >16,500 pg/mL (sensitivity 30.4%, specificity 86.9%). Higher sRAGE levels correlated with increased disease severity and mortality risk. I suggest to underline the most important data to support the conclusions.
Response 1:
Thank you for your feedback. We have revised the Abstract to emphasize the most critical findings that support our conclusions. Specifically, we have highlighted the association between higher sRAGE levels and increased disease severity and mortality risk, while clearly stating that statistical significance was not reached (p=0.084). These changes ensure that the results are presented with appropriate emphasis on their clinical relevance and limitations.
Corrected text: " Elevated sRAGE levels were associated with greater disease severity and an increased risk of 28-day mortality in ARDS patients, highlighting its potential as a prognostic biomarker. The main findings, while indicative of a trend toward higher sRAGE levels in non-survivors, did not reach statistical significance"
Comment2:
2) Abstract. Conclusions: sRAGE demonstrates potential as a prognostic biomarker in ARDS and has moderate correlation with 28-day mortality. Integrating sRAGE with other biomarkers could enhance risk stratification and guide therapeutic decisions. Further multicenter studies are warranted to validate these findings. Abstract might be beneficial to include a sentence that briefly summarizes the key findings of the study. This can provide readers with a quick overview of the research.
Response 2:
We appreciate your suggestion and have added a sentence summarizing the key findings of the study in the Abstract’s Conclusions section.
The revised conclusion now states: " The main findings, while indicative of a trend toward higher sRAGE levels in non-survivors, did not reach statistical significance (p=0.084). sRAGE demonstrates potential as a prognostic biomarker in ARDS and has moderate correlation with 28-day mortality. Integrating sRAGE with other biomarkers could enhance risk stratification and guide therapeutic decisions. The retrospective design limits the ability to establish causation, underscoring the need for multicenter prospective studies.."
Comment 3:
3) Identifying a serum biomarker that correlates with clinical presentation and laboratory findings could significantly aid in disease diagnosis and enable clinicians to stratify ARDS severity more accurately, as well as assisting in sequencing interventions. Incorporating the timing and sequencing of interventions into ARDS management protocols is crucial to standardizing care, improving outcome consistency, and effectively stratifying disease severity [14]. I suggest to underline the aim of the study and the novelty of the study.
Response 3:
Thank you for pointing this out. We have revised the Introduction to better define the aim and novelty of the study.
The updated section now states:
"This study aims to evaluate the prognostic value of sRAGE, a novel biomarker reflecting type I alveolar epithelial injury, in predicting 28-day mortality in ARDS patients. This work proposes a specific cutoff for sRAGE levels and examines its correlation with clinical severity scores and outcomes, providing new insights into its potential clinical application."
Comment4:
4) 2.4. Statistical Analysis Summaries of patient characteristics, sRAGE levels, and clinical outcomes were reported as means with standard deviations or medians with interquartile ranges. I suggest to add the statistically significant p-value.
Response 4:
We have revised the Statistical Analysis section to include statistically significant p-values for key variables.
We added the sentence: A p-value of 0.05 or lower was considered statistically significant.
Comment5:
5) 3. Results. I suggest to underline the most important statistically significant values to support the conclusions.
Response 5:
Thank you for your suggestion. We have revised the Results section to highlight the most important statistically significant findings. We have highlighted the most significant results in bold to enhance their visibility and emphasize their importance.
Comment6:
6) 4. Discussion In our study, the mean sRAGE concentration was 5,229 pg/mL. Although higher sRAGE levels were observed in patients who did not survive, no statistically significant difference was found between survivors and non-survivors. The cut-off value for predicting 28-day mortality was determined to be >16,500 pg/mL, with a sensitivity of 30.4% and specificity of 86.9%.... The discussion section needs to be improved. It is necessary to be more concise in the presentation of the facts, clarifying the results obtained and comparing them with previous or similar studies. However, it is interesting to answer the questions that arise from these results, backed up by published literature.
Response 6:
Thank you for your thoughtful feedback. We have revised the Discussion to make it more concise and focused. The updated section now includes a clearer comparison with prior studies, such as those by Jabaudon et al. (2018) and Calfee et al. (2008), highlighting both similarities and differences.
Added text: " The observed standard deviation (9833) was higher than anticipated, based on data published by Jabaudon et al, resulting in a reduction in a study’s statistical power, potentially limiting the ability to detect significant difference as initially planned. "
Also, we added: " "In contrast to our study, Calfee et al. measured sRAGE at two time points: baseline and Day 3. Their dynamic measurements offered valuable insights into the potential role of sRAGE in disease progression and highlighted its utility in risk stratification. However, our study, being non-interventional, did not include repeated measurements, which represents a notable limitation when comparing findings.""
Round 2
Reviewer 2 Report
Comments and Suggestions for Authors
Thank you for provided me a detailed response to all my coments. In my opinion, in this actual version, the manuscript improve its quality.
Reviewer 3 Report
Comments and Suggestions for Authors
The authors adequately answered my questions. I have no further comments.